# Kv7 Channels in Cyclic-Nucleotide Dependent Relaxation of Rat Intra-Pulmonary Artery

**DOI:** 10.3390/biom12030429

**Published:** 2022-03-10

**Authors:** Mohammed Al-Chawishly, Oliver Loveland, Alison M. Gurney

**Affiliations:** 1Division of Pharmacy and Optometry, Faculty of Biology Medicine and Health, University of Manchester, Core Technology Facility, 46 Grafton Street, Manchester M13 9NT, UK; mohammed.alchawishly@gmail.com (M.A.-C.); oliver.loveland@student.manchester.ac.uk (O.L.); 2Pharmacology and Toxicology Department, College of Pharmacy, Hawler Medical University, Erbil P.O. Box 178, Iraq

**Keywords:** Kv7 channel, linopirdine, XE991, sildenafil, riociguat, treprostinil, cGMP, cAMP, pulmonary artery, vasodilation

## Abstract

Pulmonary hypertension is treated with drugs that stimulate cGMP or cAMP signalling. Both nucleotides can activate Kv7 channels, leading to smooth muscle hyperpolarisation, reduced Ca^2+^ influx and relaxation. Kv7 activation by cGMP contributes to the pulmonary vasodilator action of nitric oxide, but its contribution when dilation is evoked by the atrial natriuretic peptide (ANP) sensitive guanylate cyclase, or cAMP, is unknown. Small vessel myography was used to investigate the ability of Kv7 channel blockers to interfere with pulmonary artery relaxation when cyclic nucleotide pathways were stimulated in different ways. The pan-Kv7 blockers, linopirdine and XE991, caused substantial inhibition of relaxation evoked by NO donors and ANP, as well as endothelium-dependent dilators, the guanylate cyclase stimulator, riociguat, and the phosphodiesterase-5 inhibitor, sildenafil. Maximum relaxation was reduced without a change in sensitivity. The blockers had relatively little effect on cAMP-mediated relaxation evoked by forskolin, isoprenaline or treprostinil. The Kv7.1-selective blocker, HMR1556, had no effect on cGMP or cAMP-dependent relaxation. Western blot analysis demonstrated the presence of Kv7.1 and Kv7.4 proteins, while selective activators of Kv7.1 and Kv7.4 homomeric channels, but not Kv7.5, caused pulmonary artery relaxation. It is concluded that Kv7.4 channels contribute to endothelium-dependent dilation and the effects of drugs that act by stimulating cGMP, but not cAMP, signalling.

## 1. Introduction

Drugs that stimulate cyclic nucleotide signalling in pulmonary artery smooth muscle cells (PASMCs) are the mainstay of targeted therapies used to treat pulmonary arterial hypertension (PAH). They mimic the actions of the endothelium-derived mediators, nitric oxide (NO) and prostacyclin (PGI_2_), which are depleted in pulmonary hypertension due to loss of endothelial function (e.g., [1,2]). Both mediators promote vasodilation. NO accomplishes this by stimulating soluble guanylate cyclase (sGC) and downstream cyclic guanosine monophosphate (cGMP) signalling pathways. cGMP synthesis can also be stimulated by atrial natriuretic peptide (ANP), acting through a distinct guanylate cyclase (pGC) restricted to the plasma membrane [3], but this pathway is not exploited therapeutically. PGI_2_ acts on G_s_-protein-coupled receptors on the smooth muscle cell membrane to stimulate adenylate cyclase (AC) and cyclic adenosine monophosphate (cAMP) signalling. Although these pathways in PASMCs are widely documented (e.g., [4,5]), the precise mechanisms of cyclic nucleotide action remain unresolved.

Potassium ion channels of the Kv7 family have been implicated in the regulation of pulmonary vascular tone [6,7,8,9] and can be modulated by cyclic nucleotides [10]. Intracellular cGMP activates homomeric Kv7.4 channels [11], while nitric oxide (NO) promotes the activation of homomeric Kv7.5 channels [12]. Furthermore, NO donors and the direct sGC stimulator riociguat induced K^+^ current in PASMC, hyperpolarised the cell membrane, and relaxed isolated pulmonary arteries, effects that were inhibited by Kv7 channel blockers [12]. Thus, Kv7 channel activation contributes to the pulmonary vasodilator action of NO. The role of Kv7 channels in cGMP-dependent dilation varies among systemic vessels and with the mode of cGMP generation. Thus, Kv7 channel blockers impaired NO-induced relaxation of rat aorta, but not rat renal artery, although they inhibited the relaxation of renal artery evoked by ANP [11]. It is not yet known if Kv7 channel activation in pulmonary artery requires cGMP to be produced throughout the cell by sGC or if cGMP formed under the membrane by pGC has the same effect.

Kv7 channels also contribute to cAMP-dependent dilation of systemic arteries [13,14,15,16,17,18,19], but the extent varies among different vessels [15,20] and even between vessels in the same circulation [17]. Kv7.4 was originally proposed as the target subunit, but it is now thought that Kv7.5 is needed for cAMP to activate the channels [16]. Heteromeric Kv7.4/Kv7.5 channels are probably the main mediator of cAMP effects in systemic arteries, with a preferred stoichiometry of two Kv7.4 with two Kv7.5 subunits [14,16,21]. It is not yet known if Kv7 channels play a role in the cAMP-dependent dilation of pulmonary artery.

This study investigated the contributions of Kv7 channels to cGMP- and cAMP-dependent pulmonary artery dilation. A pharmacological approach was taken to block Kv7 channels while measuring artery relaxation when cyclic nucleotide pathways were stimulated. NO donors and riociguat were used to activate sGC, ANP to activate pGC and sildenafil to inhibit phosphodiesterase 5 (PDE5) and prevent cGMP breakdown. Endothelium-derived NO was generated using the muscarinic agonist, carbachol, or the Ca^2+^ ionophore A23187. The cAMP signalling pathway was stimulated with either treprostinil or isoprenaline, which binds to and activates G_s_-coupled β_2_ receptors on PASMCs, or forskolin to stimulate AC directly [22]. The expression of Kv7 channel proteins was also investigated, because although Kv7.1, Kv7.4 and Kv7.5 mRNA transcripts are present in rat pulmonary arteries [7], it is not clear if they all translate into functional subunit proteins. The results show that Kv7 channel activation contributes substantially to cGMP-dependent dilation of pulmonary artery evoked by the NO or ANP pathways, but not cAMP-mediated dilation. Functional Kv7.1 and Kv7.4 proteins are expressed in rat pulmonary arteries, and cGMP likely acts via Kv7.4 homomers.

## 2. Materials and Methods

### 2.1. Myography

The use of animal tissues conformed to United Kingdom legislation under the Animals (Scientific Procedures) Act 1986, Amendment Regulations (SI 2012/3039). The lungs were removed rapidly from male Sprague–Dawley rats (280–360 g) euthanised by cervical dislocation and placed in ice-cold physiological salt solution (PSS) composed of (mM): NaCl 122, KCl 5, HEPES (2-[4-(2-hydroxyethyl) piperazin-1-yl] ethanesulphonic acid) 10, KH_2_PO_4_ 0.5, Na_2_HPO_4_ 0.5, MgCl_2_ 1, CaCl_2_ 1.8, glucose 5; pH 7.4 with NaOH. Intrapulmonary arteries were isolated, and rings mounted in a myograph chamber (DMT A/S, Aarhus, Denmark) under 4 mN basal tension, for isometric tension recording as previously described [6,7]. The chamber was filled with PSS at 36 °C and continuously bubbled with air. Vessels were challenged three times for 5 min each time with 50 mM KCl and then washed and allowed to recover before applying 1 μM phenylephrine to evoke sustained contraction. In some experiments, contraction was generated by raising the extracellular K^+^ concentration to 80 mM by replacing KCl in the PSS with equimolar NaCl. Once tension reached a steady level, increasing concentrations of a vasodilator or Kv7 channel activator were applied cumulatively and any change in tension recorded. Relaxation was measured as the percent reduction in amplitude of the induced tone present immediately before a drug was applied. The effects of Kv7 channel blockers were tested by adding them with the vasoconstrictor: control and treated vessels were usually studied in parallel at the same time in separate chambers.

### 2.2. Kv7 Subunit Expression

Protein was extracted from isolated pulmonary arteries, the lung and the brain. Tissue samples were homogenised in chilled RIPA lysis buffer containing 25 mM Tris-HCl (pH 7.5), 150 mM NaCl, 1 mM ethylenediamintetraacetic acid (EDTA), 1% NP-40 (*v*/*v*), 0.5% sodium deoxycholate (*w*/*v*), 1% sodium dodecyl sulphate (SDS, *w*/*v*), 1 mM phenylmethylsulphonyl fluoride (PMSF) and 1× complete, Mini, EDTA-free protease inhibitor cocktail (Roche, UK). The homogenate was centrifuged at 18,000× *g* for 15 min at 4 °C, the supernatant aspirated, and its protein content measured using the bicinchoninic acid (BCA) assay with Nanodrop 1000 spectrophotometer. Lysates were stored at −80 °C for later analysis.

Aliquots of lysate containing 25 μg of protein were diluted 4:1 with 5× Laemmli buffer, and the protein was denatured by boiling for 5 min at 95 °C. Samples were then subjected to electrophoresis using a 10% SDS-PAGE gel in a Mini-PROTEIN^®^ 3 Cell electrophoresis apparatus (Bio-Rad, Watford, Hertfordshire, UK), alongside molecular weight markers (Precision Plus Protein Dual Colour standards from Bio-Rad). The separated proteins were transferred to a polyvinylidene-fluoride (PVDF) membrane using a Trans-Blot^®^ Cell apparatus (Bio-Rad), washed 3 times with Tris-buffered saline containing 0.1% Tween-20 (TTBS), blocked for 1 h with TTBS containing 5% dried milk powder, washed again, and then incubated overnight at 4 °C with a primary antibody. Antibody binding was detected using a horseradish peroxidase (HRP)-linked secondary antibody with SuperSignal^TM^ chemiluminescent substrate (Thermo Fisher Scientific, Altrincham, Cheshire, UK) and imaged with a ChemiDoc system (Bio-Rad). Primary antibodies were obtained as follows: mouse monoclonal Kv7.1 and Kv7.4 from NeuroMab (University of California Davis), rabbit polyclonal Kv7.5 from Alomone Labs (Jerusalem, Israel), mouse monoclonal α-tubulin from Sigma-Aldrich (Poole, Dorset, UK), anti-mouse IgG/HRP from Jackson ImmunoResearch (Ely, Cambridgeshire, UK) and anti-rabbit IgG/HRP from Dako (Stockport, Cheshire, UK).

### 2.3. Sources of Drugs

Phenylephrine hydrochloride, sodium nitroprusside (SNP), ANP (127–150 rat), XE991 (10,10-bis(4-pyridinylmethyl)-9(10*H*)-anthracenone dihydrochloride), carbachol chloride, isoprenaline hydrochloride, forskolin, mefenamic acid, diclofenac sodium, levcromakalim and retigabine were sourced from Sigma-Aldrich. Linopirdine hydrochloride, HMR1556 (*N*-[(3*R*,4*S*)-3,4-Dihydro-3-hydroxy-2,2-dimethyl-6-(4,4,4-trifluorobutoxy)-2*H*-1-benzopyran-4-yl]-*N*-methylmetanesulfonamide), treprostinil, R-L3 (5-(2-Fluorophenyl)-1,3-dihydro-3-(1*H*-indol-3-ylmethyl)-1-methyl-2*H*-1,4-benzodiazepin-2-one), ML277 ((2*R*)-*N*-[4-(4-Methoxyphenyl)-2-thiazolyl]-1-[(4-methylphenyl)sulfonyl]-2-piperidinecarboxamide), ICA-069673 (*N*-(2-Chloro-5-pyrimidinyl)-3,4-difluorobenzamide), GW0742, SC-51322 and iberiotoxin were obtained from Tocris Bioscience (Bristol, UK). Sildenafil citrate was supplied by Cambridge Bioscience (Cambridge, UK), A23187 was from Calbiochem (Beeston, Nottinghamshire, UK) and glyceryl trinitrate (GTN) from Aspire Pharma Ltd. (Petersfield, UK). Depending on solubility, stock solutions were prepared as 100 or 10 mM drug in water or dimethyl sulphoxide (DMSO) and stored as frozen aliquots. DMSO was present at <0.1% in the myograph chamber.

### 2.4. Data Analysis

Data are expressed and plotted as mean ± s.e.m. of vessels from “*n*” animals. To compare concentration–response relationships in different conditions, the maximum relaxation effect (E_max_) and the drug concentration evoking 50% of the maximum effect (EC_50_) were measured. All statistical analyses employed Prism software version 7.04 (GraphPad Software, San Diego, CA, USA). The D’Agostino–Pearson omnibus and Shapiro–Wilk tests were used to assess how well data followed a normal distribution and parametric or non-parametric tests of statistical significance were applied accordingly. Two groups of data were compared using an unpaired, two-tailed *t* test. Comparison of three or more data sets employed one-way ANOVA with Dunnett’s multiple comparisons test. Two-way repeated measures ANOVA was employed to compare concentration–response curves when E_max_ or EC_50_ could not be resolved. Differences were considered statistically significant when *p* < 0.05.

## 3. Results

### 3.1. cGMP-Dependent Vasodilation

Figure 1 shows how Kv7 channel blockers influenced rat pulmonary artery relaxation mediated by vasodilators that stimulate cGMP synthesis in different ways: SNP, GTN and ANP. The relaxation of phenylephrine-contracted pulmonary arteries evoked by all three vasodilators was markedly inhibited by 1 μM linopirdine (Figure 1). Relaxation in response to SNP, GTN or ANP reached a maximum at 86 ± 3% (*n* = 9), 61 ± 2% (*n* = 6) and 89 ± 3% (*n* = 4) of the agonist-induced tone, respectively. The maxima were essentially halved by linopirdine to 54 ± 2% (*n* = 7, *p* < 0.0001) for SNP, 28 ± 2% (*n* = 6, *p* < 0.0001) for GTN and to 42 ± 8% (*n* = 4, *p* < 0.0001) for ANP. Inhibition occurred without a change in pEC_50_. The SNP pEC_50_ was 7.72 ± 0.07 in control conditions and 7.67 ± 0.07 in the presence of linopirdine, GTN values were 7.44 ± 0.07 (control) and 7.3 ± 0.1 (linopirdine) and ANP relaxed with pEC_50_ = 9.0 ± 0.1 (*n* = 4) before and 9.0 ± 0.2 (*n* = 4) after adding linopirdine. XE991 (1 μM) also suppressed the maximum relaxation evoked by SNP (to 39 ± 4%, *n* = 5, *p* < 0.0001) and GTN (to 22 ± 2%, *n* = 6, *p* < 0.0001) without affecting pEC_50_, which remained at 7.4 ± 0.2 with SNP and 7.4 ± 0.1 with GTN. In contrast to the pan-Kv7 blockers, HMR1556 (1 μM) had no significant effect on pulmonary artery relaxation evoked by these vasodilators (Figure 1).

Riociguat relaxed pulmonary arteries almost completely with pEC_50_ = 7.9 ± 0.2 (*n* = 6) in control conditions and 7.3 ± 0.2 (*n* = 6) in the presence of linopirdine (Figure 2A). In contrast to its effects on SNP, GTN and ANP, linopirdine reduced the riociguat pEC_50_ (*p* < 0.01) without affecting the maximum relaxation. Sildenafil produced a biphasic concentration–response relationship (Figure 2B); thus, single pEC_50_ and E_max_ values could not be resolved. The first phase, seen below 1 μM sildenafil, was clearly inhibited by linopirdine and XE991 (*p* < 0.001), but the inhibition was overcome at higher sildenafil concentrations.

Linopirdine and XE991 also inhibited endothelium-dependent relaxation evoked either by the muscarinic receptor agonist, carbachol (Figure 3A), or Ca^2+^ ionophore, A23187 (Figure 3B). The pattern of inhibition was similar to that seen with the NO donors SNP and GTN. The maximum relaxation to carbachol was reduced from 93 ± 3% (*n* = 6) in control conditions to 69 ± 3% (*n* = 6) in the presence of 1μM linopirdine and 64 ± 2% (*n* = 6) in the presence of 1 μM XE991. These Kv7 channel blockers also caused a small but statistically significant (*p* < 0.05) reduction in the carbachol pEC_50_, which was 6.5 ± 0.1 in control conditions, 6.1 ± 0.1 after adding linopirdine and 6.1 ± 0.06 in the presence of XE991. A23187 relaxed pulmonary arteries with pEC_50_ = 7.7 ± 0.1 and maximum at 68 ± 6% (*n* = 8) of phenylephrine-induced tone. Responses to A23187 at all concentrations were almost abolished by linopirdine and XE991 (at 1 μM). In contrast to the pan-Kv7 blockers, HMR1556 did not significantly alter the carbachol or A23187 concentration–response curves.

### 3.2. cAMP-Dependent Vasodilation

Figure 4 shows the results of similar experiments carried out with vasodilators that activate the cAMP-signalling pathway. Isoprenaline and forskolin produced concentration–relaxation curves that were not significantly altered by HMR1556 (1 μM). There was a small but significant rightward shift in both curves when experiments were performed in the presence of linopirdine (1 μM) or XE991 (1 μM), as indicated by reduced pEC_50_ values. Isoprenaline relaxed arteries with pEC_50_ = 8.1 ± 1 (*n* = 8) in control conditions, falling to 7.73 ± 0.09 (*n* = 7) in the presence of linopridine and 7.53 ± 0.07 (*n* = 11) in the presence of XE991 (*p* < 0.05). Forskolin evoked relaxation with pEC_50_ = 7.25 ± 0.04 (*n* = 6), falling to 6.98 ± 0.07 (*n* = 7) when exposed to linopirdine and 6.72 ± 0.07 (*n* = 4) in the presence of XE991 (*p* < 0.01). Neither linopirdine nor XE991 significantly altered the maximum relaxation evoked by isoprenaline (control 95 ± 6%; linopirdine 84 ± 6%; XE991 87 ± 4%) or forskolin (control 105 ± 1%; linopirdine 107 ± 2; XE991 109 ± 1%).

Linopirdine (1 μM) and XE991 (1 μM), but not HMR1556 (1 μM), inhibited pulmonary artery relaxation evoked by treprostinil (Figure 5A). This inhibition had a different profile, however, from the inhibition of isoprenaline and forskolin. Both Kv7 blockers markedly reduced the maximum treprostinil-induced relaxation: 80 ± 6% (*n* = 7) in control conditions, 48 ± 8% (*n* = 6; *p* < 0.01) in the presence of linopirdine and 36 ± 5% (*n* = 6; *p* < 0.001) in the presence of XE991. This occurred without a change in the treprostinil pEC_50_, which was 7.79 ± 0.04 in control conditions, 7.6 ± 0.03 in the presence of linopirdine and 7.7 ± 0.09 in the presence of XE991. Treprostinil is known to bind to and activate peroxisome proliferator-activated receptors (PPARs) independently of cAMP [23]. As the PPARβ activator, GW0742, causes concentration-dependent pulmonary artery relaxation [24], we investigated how it might be impacted by Kv7 channel blockers. As shown in Figure 5B, XE991 did not significantly alter the GW0742 concentration–relaxation curve. Relaxation occurred with pEC_50_ = 5.78 ± 0.06 (*n* = 8) in control conditions and 5.3 ± 0.07 (*n* = 8) in the presence of XE991. A maximum relaxation of 102 ± 2% (*n* = 8) was achieved both before and after applying XE991.

A substantial part of the treprostinil-induced relaxation of rat pulmonary arteries has been found to require the presence of an intact endothelium and was blocked by inhibiting the endothelial NO synthase [25]. Consistent with this, the efficacy of treprostinil was reduced when tested in the presence of L-NAME (200 μM) to block NO synthase (Figure 5C), leaving a maximum relaxation of only 31 ± 6% (*n* = 8, *p* < 0.0001). It was reduced further to 16 ± 2% (*n* = 8, *p* < 0.01) when XE991 (1 μM) was added. The treprostinil pEC_50_ was reduced to 7.3 ± 0.2 (*n* = 8, *p* < 0.001) by L-NAME, but was not further affected by XE991 (7.3 ± 0.1, *n* = 8). Treprostinil lacks receptor selectivity. While its activation of IP, DP_1_ and EP_2_ receptors leads to stimulation of AC and cAMP synthesis, it can also bind to prostanoid receptors that evoke rat pulmonary artery contraction and counteract relaxation [26,27], explaining the loss of relaxation at treprostinil concentrations ≥1 μM (Figure 5A). While treprostinil has a low affinity for the EP_3_ receptor (K_i_ > 2μM), thought to be the main mediator of contraction, it binds the EP_1_ receptor (K_i_ = 285 nM) with only 10-fold lower affinity than IP [26]. When applied at a concentration expected to inhibit both EP_1_ and EP_3_ receptors, SC-51322 (10 μM) enhanced the treprostinil-induced relaxation (maximum = 42 ± 8%, *n* = 6) measured in the presence of L-NAME (Figure 5D). In these conditions, XE991 (1 μM) failed to impede the relaxation (Figure 5D).

### 3.3. Kv7 Channel Pharmacology

The Kv7.1 selective activators R-L3 [28] and ML277 [29] both evoked concentration-dependent relaxation of rat pulmonary arteries (Figure 6A(i),A(ii),B,C). R-L3 produced up to 96 ± 2% (*n* = 12) inhibition of phenylephrine-induced tone with pEC_50_ = 6.62 ± 0.04 (EC_50_ = 240 nM). ML277 relaxed phenylephrine-contracted arteries up to a maximum of 84 ± 6% (*n* = 8) with pEC_50_ = 6.74 ± 0.02 (EC_50_ = 182 nM). The effect of K^+^ channel activation on membrane potential, and hence vascular tone, can be prevented by depleting the transmembrane K^+^ gradient. This was performed by raising the extracellular K^+^ concentration to 80 mM, which causes depolarisation and evokes contraction. R-L3 and ML277 were less effective at relaxing arteries contracted with 80 mM K^+^, compared with phenylephrine, but their effects were not abolished (Figure 6B,C).

Non-steroidal, anti-inflammatory fenamate drugs interact with Kv7 channels in an isoform-specific manner. Mefenamic acid (MFA), an activator of the Kv7.1 isoform [30,31], relaxed phenylephrine-contracted arteries with pEC_50_ = 4.6 ± 0.1 (*n* = 5; EC_50_ = 25 μM) and 99 ± 5% maximum (Figure 6A(iii),D). Diclofenac does not affect the Kv7.1 channel, but it activates Kv7.4 and inhibits Kv7.5 [32,33]. Diclofenac caused almost complete relaxation of pulmonary arteries (Figure 6A(iv)) with the concentration dependence shown in Figure 6D. The maximum response was not always clear but taking the response at the highest diclofenac concentration as a rough maximum gave pEC_50_ = 3.68 ± 0.05 (*n* = 11; EC_50_ = 209 μM). ICA-069673 was originally identified as a potent Kv7.2/Kv7.3 activator that is inactive at Kv7.1 channels [34], but at micromolar concentrations, it also activates Kv7.4 channels expressed in a vascular smooth muscle cell line [35]. ICA-069673 relaxed pulmonary arteries in a concentration-dependent manner with pEC_50_ = 5.21 ± 0.06 (*n* = 4; EC_50_ = 6.2 μM) and 104 ± 1% maximum (Figure 6A(v),E). In contrast, ICA-069673 had little effect on arteries contracted with 80 mM K^+^ (Figure 6E). 

HMR1556 (1 μM) shifted the RL-3 (Figure 6B) and ML277 (Figure 6C) concentration–response curves to the right, increasing the pEC_50_ values to 6.15 ± 0.06 (*n* = 6; *p* < 0.001) and 6.46 ± 0.08 (*n* = 8; *p* = 0.005), respectively, without changing the maximum relaxation. In contrast, HMR1556 had no effect on the ICA-069673 concentration–response curve (Figure 6E). It also failed to change the concentration–relaxation curves generated by the Kv7.2-Kv7.4 activator, retigabine (Figure 6F), or the K_ATP_ channel activator, levcromakalim (Figure 6G). As BK_Ca_ channels have been proposed to mediate cyclic nucleotide-dependent dilation of pulmonary arteries [36,37], we tested the ability of iberiotoxin, a selective BK_Ca_-channel blocker [38], to interfere with pulmonary artery relaxation mediated by cGMP or cAMP. The toxin had no effect on relaxation responses to either sildenafil (Figure 6H) or isoprenaline (Figure 6I).

### 3.4. Kv7 Channel Expression

Western blot analysis confirmed the expression of Kv7.1 and Kv7.4 proteins in pulmonary arteries, but not Kv7.5. The expression of Kv7 subunits was investigated initially in whole lung (Figure 7A) by using brain tissue for positive controls, because it expresses all three subunits [39,40] and it is a reliable source of protein, and α-tubulin as a loading control. Brain tissue expressed all three Kv7 subunits. Bands running near the molecular weights predicted for Kv7.1 (75 kDa), Kv7.4 (77 kDa) and Kv7.5 (102 kDa) were clearly identified with the relevant antibodies. In lung samples, bands of the correct molecular weight stained positively with antibodies directed against Kv7.1 and Kv7.4. The lung showed stronger expression of Kv7.1 than the brain, despite weaker α-tubulin staining. In contrast, despite a strong Kv7.5 band from the brain, no corresponding band could be detected in lysates from the lung. Figure 7B demonstrates that Kv7.1 and Kv7.4 proteins were expressed in arteries isolated from the lung. Conversely, the artery lysate did not give rise to a band compatible with Kv7.5 expression.

## 4. Discussion

The results of this study suggest that Kv7 channels contribute substantially to pulmonary vasodilation mediated by the cGMP-signalling pathway but have a lesser role in cAMP signalling. Pharmacological and expression profiling further suggests that Kv7.4 is the isoform most likely to act as the functional target of cGMP in rat pulmonary arteries.

### 4.1. cGMP Signalling

The broad-spectrum Kv7 channel blockers, linopirdine and XE991, were recently shown to impair pulmonary artery relaxation induced by NO donors and riociguat [12]. Here, substantial inhibition was found at a 10-fold lower concentration of the Kv7 blockers, which is sufficient to inhibit Kv7 current in rat PASMCs without affecting other K currents and lower than the concentrations needed to interact with other types of ion channels [7]. They caused marked inhibition of five vasodilators that act on smooth muscle to raise PASMC cGMP levels (SNP, GTN, ANP, riociguat and sildenafil). They also suppressed the effects of two vasodilators (carbachol and A23187) that raise PASMC cGMP by stimulating endothelial release of NO. The blockers were particularly effective against A23187, abolishing its effect. This might be because A23187 also releases an endothelium-dependent contracting factor [41], evident as a loss of relaxation at higher concentrations, which would be unmasked when NO action is blocked.

The pattern of inhibition depended on the method of raising cGMP. The Kv7 blockers suppressed maximum responses to SNP, GTN, ANP and endothelium-derived NO, suggesting that a key pathway required to generate full relaxation had been removed. The results support the finding of Mondéjar-Parreño and colleagues [12] that NO donors enhance Kv7 channel currents in PASMC, which hyperpolarises the cell membrane, leading to reduced Ca^2+^ influx and muscle relaxation. The pulmonary artery resembles the aorta (at least in rat), where relaxations induced by activating the pGC with ANP or the sGC with NO donors were both impaired by linopirdine, but it differs from the renal artery where only the pGC was linked to Kv7 channels [11]. Multiple pathways downstream of cGMP could contribute to smooth muscle relaxation [42], but the reduced maximum relaxation caused by Kv7 blockers means that alternative pathways are unable to compensate when there is a deficit in Kv7 channel activity.

The clear inhibition of sildenafil-induced relaxation by Kv7 blockers at sub-micromolar sildenafil concentrations was overcome at higher sildenafil concentrations. This may reflect a lack of sildenafil selectivity at higher concentrations, suggested by its biphasic concentration–response relationship. Deviation from the classical sigmoidal shape has been noted before for sildenafil [43]. Relaxation at nanomolar concentrations is due to PDE5 inhibition by sildenafil, which occurs with an EC_50_ of 2-4 nM [44], while relaxation at higher concentrations has been attributed to inhibition of cAMP-specific phosphodiesterases [43]. Thus, the inhibition by linopirdine and XE991 seen below 1 μM sildenafil concurs with their inhibition of NO- and ANP-induced relaxation. Sildenafil may be able to overcome a Kv7 channel block at higher concentrations by stimulating cAMP-mediated relaxation, which was relatively insensitive to Kv7 blockers.

While Kv7 blockers inhibited relaxation to riociguat, they increased its EC_50_ rather than reducing the maximum relaxation. This differential effect of the blockers on relaxation induced by riociguat versus NO donors is also apparent in the concentration–response curves reported by Mondéjar-Parreño et al. [12] but was not commented on by the authors. Thus, Kv7 blockers reduce the sensitivity to riociguat but do not prevent its full effect. The block can be overcome by increasing the riociguat concentration. The result is surprising, because riociguat acts at the same site on sGC as NO, to both enhance NO action and stimulate enzyme activity independently of NO [45]. Its ability to stimulate sGC depends on a reduced haem domain in the enzyme, which can be oxidised by ODQ [46] with IC_50_ ~20 nM [47]. At a 500-fold higher concentration, ODQ prevented relaxation to riociguat at up to 100 nM, mirroring its effect on riociguat stimulation of sGC at the same concentration [48]. Riociguat relaxation therefore required its interaction with reduced haem, but the relief of ODQ block at higher riociguat concentrations suggests that additional signalling molecules could be recruited as the concentration rises. This may explain why riociguat can activate relaxant pathways that are not accessible when cGMP levels are increased by NO or ANP.

### 4.2. cAMP Signalling

Linopirdine and XE991 reduced the potencies of isoprenaline and forskolin to relax pulmonary arteries without impairing the maximum relaxation. Such inhibition is consistent with the known ability of cAMP to enhance Kv7 channel activity, but the effect was small in comparison to the rat renal artery, where linopirdine and Kv7.4 gene silencing caused a ≥10-fold increase in the concentration dependence of isoprenaline and forskolin evoked relaxation [13]. Downregulation of Kv7.4 would have reduced the expression of heteromeric Kv7.4/Kv7.5 channels, thought to be the target of cAMP [10,16]. The inhibition of pulmonary artery relaxation was also small compared to the drugs’ effects on cGMP-mediated relaxation. Thus, the Kv7 channels play a relatively minor role in cAMP-mediated relaxation of pulmonary arteries, especially when compared with cGMP-dependent relaxation. cAMP-mediated relaxation may involve a cGMP component due to cyclic nucleotide cross talk [49]. Thus, the limited inhibition of isoprenaline or forskolin by Kv7 blockers might reflect their effect on the cGMP pathway. Whatever the mechanism, the inhibition by Kv7 channel blockers was overcome at higher isoprenaline and forskolin concentrations; thus, alternative cAMP signalling pathways maintain relaxation capacity when there is a Kv7 channel deficit.

At first glance, the large effect of Kv7 channel blockers on treprostinil relaxation conflicts with their effects on isoprenaline and forskolin and is more similar to the inhibition of cGMP-meditated relaxation. IP receptors are present on endothelial cells, and their activation stimulates NO synthesis (e.g., [50]). Thus, as previously reported [25], treprostinil relaxation was partly endothelium dependent and reduced by L-NAME. Blocking endothelium-dependent relaxation did not, however, prevent XE991 from suppressing the maximum treprostinil relaxation. Therefore, NO and cGMP-dependent mechanisms cannot fully explain the strong inhibition by Kv7 channel blockers. Rather, the finding that XE991 inhibition was blocked by SC-51322 implies that it was at least in part due to potentiation of an opposing contractile action of treprostinil on EP_1_ and/or EP_3_ receptors. Despite a low affinity of treprostinil for EP_3_ receptors [26], they appear to contribute to its effects on rat pulmonary artery [27]. Treprostinil has a higher affinity for EP_1_ receptors. They do not contribute to contraction in the human pulmonary artery [51,52], but their role in rats is not clear.

### 4.3. Kv7 Channel Subtypes

Rat pulmonary arteries expressed Kv7.1 protein and responded with concentration-dependent relaxation to three activators of Kv7.1 (R-L3, ML277 and mefenamic acid). ML277 and R-L3 both relaxed the rat pulmonary artery with EC_50_ values < 250 nM, and the effects were antagonised by HMR1556 at 1μM. Importantly, HMR1556 did not interfere with relaxation evoked by the Kv7.4 channel activator ICA-069673 [35], the Kv7.2-7.5 activator retigabine [53] or the K_ATP_ channel activator, levcromakalim, indicating specificity for the Kv7.1 activators. Consistent with K^+^ channel activation as a mechanism of relaxation, R-L3 and ML277 were both less effective when tested on pulmonary arteries contracted with 80 mM K^+^ instead of phenylephrine. Nevertheless, substantial relaxation remained in those conditions; thus, part of the actions of both drugs must be unrelated to Kv7.1 channels. The selectivity of Kv7.1 modulators remains to be widely tested, but the relaxation of rat [8] and mouse [54] mesenteric arteries by R-L3 or ML277 did not employ Kv7.1 channel activation. It was not prevented by HMR1556 at 10 μM and persisted in arteries from Kv7.1-deficient mice [54]. Despite its off-target effects, the selective inhibition by HMR1556 of relaxation induced by Kv7.1 activators in pulmonary artery suggests that, in contrast to mesenteric arteries, they do express functional Kv7.1 channels. These channels did not, however, contribute to cGMP- or cAMP-dependent relaxations, which were insensitive to HMR1556.

Kv7.5 protein could not be detected either in whole lung or isolated pulmonary arteries, despite the Kv7.5 mRNA transcript being present in rat PASMCs [7]. It was, however, present in samples of the brain where its expression is well established [39,55]. This finding conflicts with two earlier reports of Kv7.5 protein expression in the pulmonary artery [56,57]. The reason for the discrepancy is uncertain, but no information was provided on the antibodies used in one study [56] and the 75k Da protein detected in the other [57] is smaller than predicted (102 kDa) for Kv7.5. As cAMP activates Kv7.5-containing channels, but not homomeric Kv7.4 channels [16], the apparent lack of Kv7.5 expression in the pulmonary artery is consistent with the poor inhibition of cAMP-dependent dilation by Kv7 blockers.

In contrast, Kv7.4 protein was robustly expressed in the pulmonary artery. Moreover, the Kv7.4 activators, ICA-069673 and diclofenac, evoked artery relaxation over a concentration range shown to enhance Kv7.4 currents [33,35]. ICA-069673 was identified as a selective Kv7 activator, lacking effects on a range of other ion channels [34]. It also shows subtype selectivity within the Kv7 family, robustly enhancing homomeric Kv7.4 currents at 10 μM, while having no effect on Kv7.5 below 100 μM [35,58] and little effect on Kv7.1 current [34,58]. This compares with almost complete relaxation of pulmonary arteries at 10 μM. Importantly, ICA-069673 had almost no effect on arteries contracted with 80 mM K^+^, consistent with K-channel activation as the main cause of relaxation. The diclofenac EC_50_ for relaxation (~200 μM) is close to its EC_50_ for activating Kv7.4 (102 μM [33]). Diclofenac has also been shown to have no effect on currents mediated by Kv7.1 channels [32] and to inhibit Kv7.5 currents with EC_50_ ~ 20 μM, while having little effect on heteromeric Kv7.4/7.5 channels [33]. Although diclofenac can discriminate between members of the Kv7 channel family, as with all fenamate non-steroidal anti-inflammatory drugs, it can interact with other ion channels and cyclooxygenase enzymes. Nevertheless, when considered together, the results point to homomeric Kv7.4 channels as the candidate for mediating cGMP-dependent relaxation.

### 4.4. Exclusion of Other K Channels

Both cAMP and cGMP have been proposed to relax pulmonary arteries by activating large-conductance Ca^2+^-activated K (BK_Ca_) channels in PASMCs [36,37,59], an idea that seems to have become dogma [60,61]. The main evidence for such a role is the inhibition of relaxation by charybdotoxin [36,59], which has limited selectivity for BK_Ca_ channels. It can inhibit other Ca^2+^-activated channels, as well as Kv1.2 and Kv1.3 channels [62]. Moreover, around the time that its effect on pulmonary artery relaxation was described, some sources of the toxin were contaminated with agitoxins, which potently block some Kv channels [62,63]. The potent and more selective BK_Ca_ channel blocker, iberiotoxin [64], was recently reported to have no effect on PASMC K^+^ current or artery relaxation evoked by NO donors [12]. We additionally found that it had no effect on relaxation evoked by sildenafil or isoprenaline. BK_Ca_ channels are therefore not involved in cGMP- or cAMP-dependent pulmonary vasodilation.

### 4.5. Implications for PAH

Kv7.1 and Kv7.4 subunits are both downregulated in the lungs of patients [65] and animal models [9,12,56] with PAH. This has implications for the effectiveness of cGMP-dependent vasodilators used to treat the disease. It would limit their ability to dilate pulmonary arteries and possibly interfere with effects on cell proliferation. The physiological role of Kv7.1 in PASMCs is not known, but Kv7.4 channels help to maintain a negative membrane potential and limit Ca^2+^ influx, which promotes vasodilation [6,7] and prevents cell proliferation [66]; thus, their loss may contribute to the development of PAH. Nevertheless, despite reduced Kv7.4 expression, Kv7 activators retain the ability to dilate pulmonary arteries in PAH [9,57,67] and may even be more effective [57]. Importantly, when administered orally, the Kv7 activator, flupirtine, inhibited the development of PAH in rodents exposed to chronic hypoxia and reversed established PAH in mice overexpressing the serotonin transporter [9,67]. Kv7.4 channels therefore have potential as a target for the development of new drug treatments. As Kv7 activators act downstream of cGMP, they could be more effective, or act synergistically with cGMP-dependent dilators. In addition, the more prominent role of Kv7.4 homomers in pulmonary arteries compared with the systemic circulation suggests it may be possible to identify compounds with selectivity for the pulmonary circulation.

### 4.6. Limitations of the Study

The main limitation is the reliance on drugs to interfere with cellular pathways, especially where selectivity has not been widely tested. Although gene silencing could provide more specific knockout of channel subunits from the artery, in our hands it does not suppress expression reproducibly or reliably. To minimize the problem, drug concentrations were selected to avoid known off-target interactions. While no Kv7 modulator is specific enough to rely on by itself, conclusions were drawn from the pattern of effects of multiple drugs with similar or opposing actions on different Kv7 channels. While the evidence shows that cAMP does not act through Kv7 channels in pulmonary artery, its effects on the channels have not been investigated directly using electrophysiological techniques.

## 5. Conclusions

The results of this study strongly support a role for Kv7 channels, but not BK_Ca_ channels, in mediating cGMP-dependent pulmonary vasodilation. Whether resulting from endothelium-released NO, ANP, NO donors, guanylate cyclase stimulators, or PDE5 inhibition, relaxation was inhibited by Kv7 channel blockers. Any contribution of Kv7 channels to cAMP-dependent pulmonary vasodilation was, however, small, with other mechanisms dominating. Kv7.4 is the best candidate for mediating cGMP relaxation, because: (1) blocking Kv7.1 channels did not affect relaxation; (2) Kv7.5 channels are poorly expressed in PASMC; (3) drugs that directly activate Kv7.4, but not Kv7.5, produced robust pulmonary artery relaxation.

The effects of therapeutic drugs that depend on cGMP for their action may be limited by downregulation of Kv7.4 channels in PAH. As Kv7 channel activators can reduce PAH, they offer an additional target for drug development.

## Figures and Tables

**Figure 1 biomolecules-12-00429-f001:**
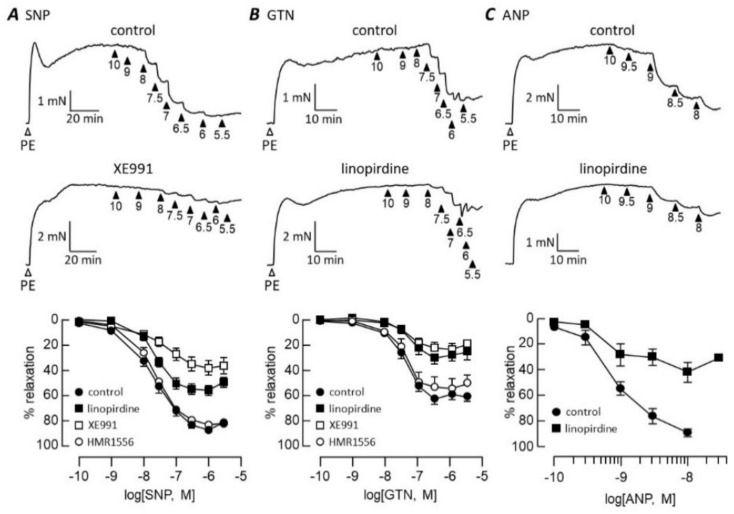
Kv7 channel blockers inhibit NO- and ANP-evoked relaxation. Traces illustrate pulmonary artery contraction to 1 μM phenylephrine (PE), followed by relaxation responses to increasing concentrations of SNP (**A**), GTN (**B**) or ANP (**C**) applied cumulatively in control conditions (upper traces) and after incubation with 1 μM XE991 or linopirdine (lower traces). Arrowheads indicate -log[dilator]. Concentration–response curves (bottom panel) are plotted in control conditions and after incubation with 1 μM Kv7 blocker as indicated. SNP *n* = 5 (XE991), 9 (control) or 7; GTN *n* = 6; ANP *n* = 4.

**Figure 2 biomolecules-12-00429-f002:**
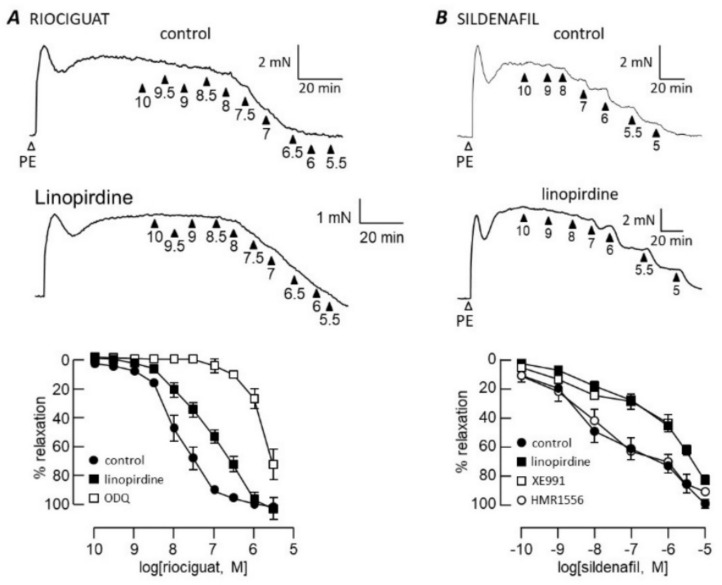
Kv7 channel blockers inhibit cGMP-dependent relaxation. Traces illustrate pulmonary artery contraction to 1 μM phenylephrine (PE), followed by relaxation responses to increasing concentrations of riociguat (**A**) or sildenafil (**B**) applied cumulatively in control conditions (upper traces) and after incubation with 1 μM linopirdine (lower traces). Arrowheads indicate −log[dilator]. Concentration–response curves (bottom panel) are plotted in control conditions and after incubation with 1 μM Kv7 blocker as indicated. Riociguat *n* = 5 (control and linopirdine) or 3 (ODQ); sildenafil *n* = 4 (HMR1556 and XE991), 10 (control) or 11 (linopirdine).

**Figure 3 biomolecules-12-00429-f003:**
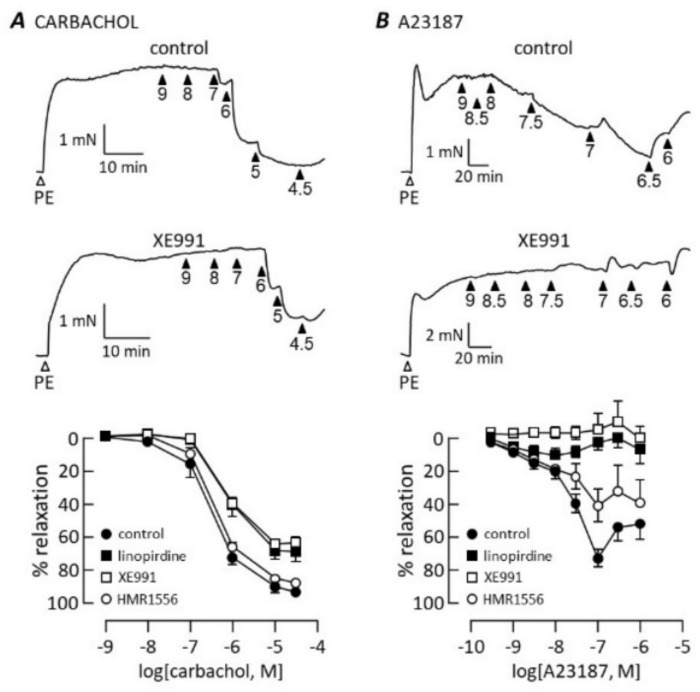
Kv7 channel blockers inhibit endothelium-dependent vasodilation. Traces illustrate contraction to 1 μM phenylephrine (PE) followed by relaxation responses to carbachol (**A**) or A23187 (**B**), applied cumulatively at the concentrations indicated by the arrowheads (−log[dilator]), in control conditions (upper traces) and after incubation with 1μM XE991 (lower traces). Graphs show concentration–response curves to carbachol (**A**, *n* = 6) and A23187 (**B**, *n* = 8–9) in control conditions and after incubation with 1 μM XE991, linopirdine or HMR1556 as indicated.

**Figure 4 biomolecules-12-00429-f004:**
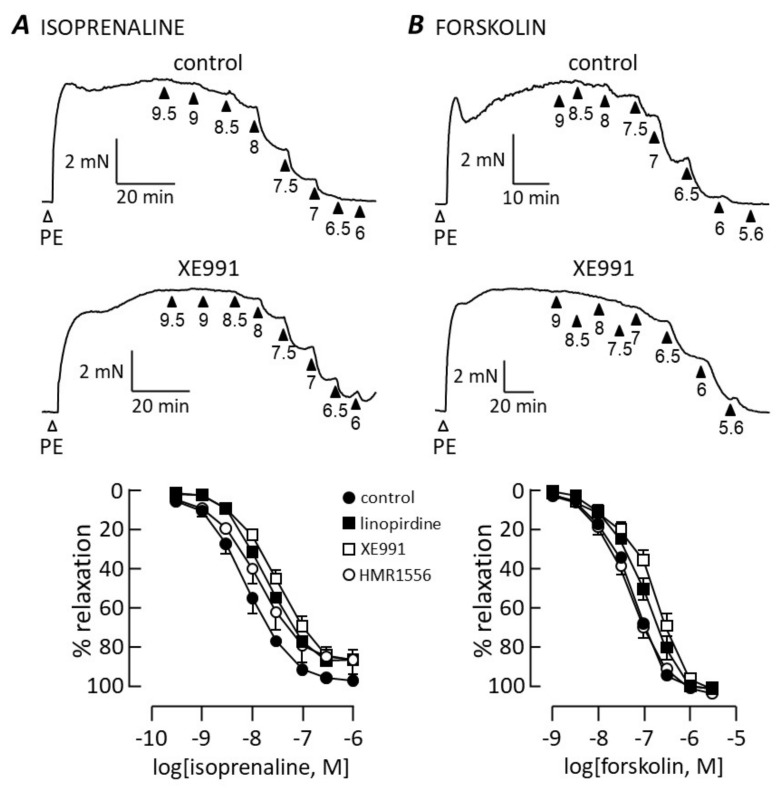
Relaxation mediated by cAMP is resistant to Kv7 channel blockers. Traces illustrate contraction to 1 μM phenylephrine (PE) followed by relaxation to isoprenaline (**A**) or forskolin (**B**) added cumulatively at the concentrations (−log[dilator]) indicated by the arrowheads. Upper traces show controls and lower traces in the presence of 1 μM XE991. Concentration–response relationships are compared in control conditions and after adding a Kv7 channel blocker (1 μM) as indicated, for isoprenaline (**A**; control *n* = 8, linopirdine *n* = 7, XE991 *n* = 11, HMR1556 *n* = 5) and forskolin (**B**; control *n* = 6, linopirdine *n* = 4, XE991 *n* = 4, HMR1556 *n* = 6).

**Figure 5 biomolecules-12-00429-f005:**
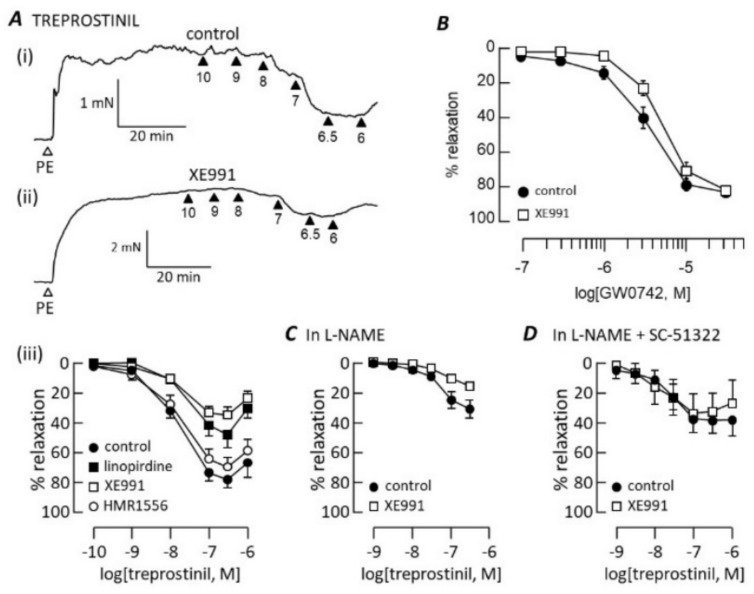
Kv7 channel blockers inhibit treprostinil-induced relaxation. Traces illustrate contraction to 1 μM phenylephrine (PE) followed by relaxation after cumulative addition of 0.1 nM–1 μM treprostinil in control conditions (**A**(**i**)) and after adding 1 μM XE991 (**A**(**ii**)). Arrowheads indicate -log[treprostinil]. Concentration–relaxation curves (**A**(**iii**)) are compared in control conditions and after adding a Kv7 channel blocker (1 μM), as indicated (*n* = 6 in each condition). (**B**) Concentration–response curves to GW0742 in the absence and presence of 1 μM XE991 (*n* = 8). Treprostinil concentration–response curves are compared in the absence and presence of 1 μM XE991 after blocking NO synthase with 100 μM L-NAME (**C**; *n* = 8) and with additional block of EP receptors with 10 μM SC-51322 (**D**; *n* = 5).

**Figure 6 biomolecules-12-00429-f006:**
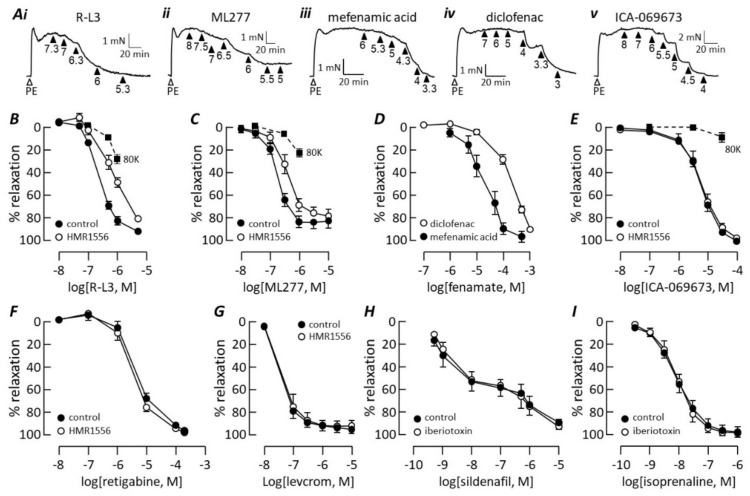
Pharmacology of K channels mediating pulmonary artery relaxation. (**A**) Traces illustrate contractile responses to 1 μM phenylephrine followed by relaxation evoked by cumulative addition of R-L3 (**A**(**i**)), ML277 (**A**(**ii**)), mefenamic acid (**A**(**iii**)), diclofenac (**A**(**iv**)) or ICA-069673 (**A**(**v**)). Concentrations (as −log[drug]) are indicated below the arrowheads. (**B**) RL-3 concentration–response curves measured in the absence (*n* = 12) and presence (*n* = 6) of 1 μM HMR1556 or when vessels were contracted with 8o mM K^+^ (8oK, *n* = 5). (**C**) ML277 concentration–response curves in the absence (*n* = 8) and presence (*n* = 8) of 1 μM HMR1556 or when vessels were contracted with 8oK (*n* = 5). (**D**) Concentration-response curves to mefenamic acid (*n* = 5) and diclofenac (*n* = 11). (**E**) ICA-069673 concentration–response curves in the absence (*n* = 4) and presence (*n* = 4) of 1 μM HMR1556 or when vessels were contracted with 8oK (*n* = 6). (**F**) Retigabine concentration–response curves measured in the absence (*n* = 9) and presence (*n* = 7) of 10 μM HMR1556. (**G**) Levcromakalim concentration–response curves in the absence (*n* = 5) and presence (*n* = 4) of 1 μM HMR1556. (**H**) Sildenafil concentration–response curves measured in the absence (*n* = 7) and presence (*n* = 4) of 100 nM iberiotoxin. (**I**) Isoprenaline concentration–response curves measured in the absence (*n* = 8) and presence (*n* = 4) of 100 nM iberiotoxin.

**Figure 7 biomolecules-12-00429-f007:**
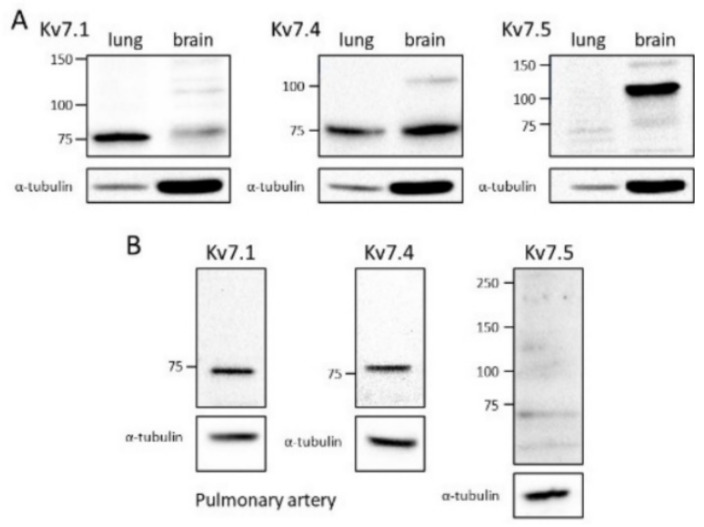
Kv7 channel protein expression in rat pulmonary arteries. Immunoblots of Kv7 channel α-subunit and α-tubulin proteins, using lysates isolated from whole rat lung or brain (**A**) or isolated rat pulmonary arteries (**B**). Each blot is typical of experiments repeated on tissues from at least 4 rats. Wells were injected with lysate containing 25 μg of protein. Numbers to the left of each blot indicate the positions of molecular weight markers (kDa). Each gel was cut to stain separately for α-tubulin (50 kD), used as a loading control.

## Data Availability

Not applicable.

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
