# Peer review of "Kv7 Channels in Cyclic-Nucleotide Dependent Relaxation of Rat Intra-Pulmonary Artery"

_biomolecules, 2022, doi:10.3390/biom12030429_

Round 1
Reviewer 1 Report
The study by Al-Chawishly et al aims to explore the contributions of Kv7 channels to cGMP- and cAMP-dependent pulmonary vasodilation. By using vascular reactivity assays, authors found that pulmonary vasodilation induced by drugs stimulating the cGMP pathway, is reduced by Kv7 channel blockers. In contrast, pulmonary arterial relaxation to dugs acting through the cAMP pathway was relatively resistant to Kv7 channel inhibitors. This is a well-designed and performed study, but the main concern is that the novelty of the study is limited, as previous studies have already described the involvement of Kv7 channels in cGMP-dependent pulmonary vasodilation. However, in my opinion the results are interesting as they reaffirm the role of Kv7 channels (instead of other previously involved such as BKCa) in pulmonary relaxation via cGMP and suggest a poor role in the cAMP pathway. I have several comments and suggestions:
- The work essentially shows vascular reactivity data. Electrophysiological data would be needed to support the findings. In this sense, it is recommended to include data confirming that drugs stimulating the cGMP but not those stimulating cAMP, enhance Kv7 currents in myocytes of pulmonary arteries.
- Figure 3 shows that pan-Kv7 blockers abolished the relaxation to the Ca2+ ionophore A23187 while only modestly attenuating that induced by carbachol. Given that both are supposed to stimulate endothelium-dependent vasodilation, how do the authors interpret this finding?
- Figure 7 shows one representative western blot of Kv7 channel protein expression in lung and rat pulmonary arteries. Their data suggest a poor expression of Kv7.5 compared to Kv7.1 and Kv7.4. Nevertheless, this is just one experiment. In order to be representative samples from at least 3 different animals should be analysed to confirm the preliminary data.
- According to the authors' data, Kv7 channels appear to play a relatively minor role in cAMP-mediated relaxation of pulmonary arteries. This is in contradiction to many studies in systemic arteries showing a role for these channels in cAMP-induced relaxation. In the discussion it is suggested that the mechanism for these differences could be due to a lower expression of Kv7.5. However, there is much evidence in the literature showing that cAMP-induced vascular relaxation mainly involves Kv7.4 channels. Do the authors have any alternative explanation for their data in this regard?
- As the authors state pulmonary hypertension is treated with drugs that stimulate cGMP or cAMP signalling. It is therefore strongly recommended that authors discuss their data in the context of pulmonary hypertension in line with this special Issue “Role of Ion Channels Signaling Pathways in the Development of Pulmonary Arterial Hypertension”. In this regard, there is evidence in the literature on the pathophysiological and pharmacological role for Kv7 channels in the pulmonary circulation (i.e. PMID: 33306938, PMID: 32829658, PMID: 25361569, PMID: 19508393).
Minor:
In the abstract line 19 “The Kv1-selective blocker, HMR1556, …” should be modified by “The Kv7.1-selective blocker, HMR1556, …”
Reviewer 2 Report
SUMMARY
This study uses ex vivo analysis of rat tissue to assess the role of Kv7 channels in mediating cGMP-dependent pulmonary vasodilation. There are a few comments I would like the authors to address:
MAJOR
For all drugs used, are the drugs specific to the molecule/pathway/channel being studied? If so, the data and/or references to prove this should be clearly stated. If not, there should be explicit admittance of this at first mention and throughout.
Two-way ANOVA is mentioned in the Results, but not the Statistics section. It must be clear throughout, which stats tests are being used.
Do the n numbers refer to number of mice or number of arteries or number of tests? Please make this clear.
MINOR
It is not entirely clear why this study could be useful for patients.
The Introduction is lengthy and may benefit by shortening.
Why was brain chosen as a control for lung in the Western Blotting?
A separate section on study limitations could be useful in the Discussion.
Round 2
Reviewer 1 Report
Authors have addressed most of my concerns.
I only have minor comments:
1. I understand that, although the electrophysiological data I suggested are highly recommended to provide more robustness to the myography data, they require more than the 10-day period the journal allows for manuscript review. Maybe this could be included in the "Limitations of the study section"
2. The study cited in Line 641 (11) did not involve pulmonary arteries but systemic arteries, please correct it
Author Response
A sentence has been added to the limitations section at lines 521-23. It reads "While the evidence shows that cAMP does not act through Kv7 channels in pulmonary artery, its effects on the channels have not been investigated directly using electrophysiological techniques."
We found an incorrect citation of reference 11 at line 493, which has been corrected. It must have happened when the references were renumbered during the first revision. We trust that is the one you referred to as line 641 is within a reference.